# Electron Microscopic Confirmation of Anisotropic Pore Characteristics for ECMO Membranes Theoretically Validating the Risk of SARS-CoV-2 Permeation

**DOI:** 10.3390/membranes11070529

**Published:** 2021-07-14

**Authors:** Makoto Fukuda, Tomoya Furuya, Kazunori Sadano, Asako Tokumine, Tomohiro Mori, Hitoshi Saomoto, Kiyotaka Sakai

**Affiliations:** 1Department of Biomedical Engineering, Kindai University, 930 Nishimitani, Kinokawa-city, Wakayama 649-6493, Japan; to1mo1ya7.f@gmail.com (T.F.); 1718360065r@waka.kindai.ac.jp (K.S.); tokumine@waka.kindai.ac.jp (A.T.); 2Industrial Technology Center of Wakayama Prefecture, 60 Ogura, Wakayama-city, Wakayama 649-6261, Japan; tomohiro_mori@wakayama-kg.jp (T.M.); saomoto@wakayama-kg.jp (H.S.); 3Department of Chemical Engineering, Waseda University, 3-4-1 Okubo, Shinjuku-ku, Tokyo 169-8555, Japan; kisakai@waseda.jp

**Keywords:** extracorporeal membrane oxygenator (ECMO), artificial lung, extracorporeal membrane oxygenation (ECMO), COVID-19, plasma leakage, ECMO infection, polypropylene (PP), silicone layer, polymethylpentene (PMP)

## Abstract

The objective of this study is to clarify the pore structure of ECMO membranes by using our approach and theoretically validate the risk of SARS-CoV-2 permeation. There has not been any direct evidence for SARS-CoV-2 leakage through the membrane in ECMO support for critically ill COVID-19 patients. The precise pore structure of recent membranes was elucidated by direct microscopic observation for the first time. The three types of membranes, polypropylene, polypropylene coated with thin silicone layer, and polymethylpentene (PMP), have unique pore structures, and the pore structures on the inner and outer surfaces of the membranes are completely different anisotropic structures. From these data, the partition coefficients and intramembrane diffusion coefficients of SARS-CoV-2 were quantified using the membrane transport model. Therefore, SARS-CoV-2 may permeate the membrane wall with the plasma filtration flow or wet lung. The risk of SARS-CoV-2 permeation is completely different due to each anisotropic pore structure. We theoretically demonstrate that SARS-CoV-2 is highly likely to permeate the membrane transporting from the patient’s blood to the gas side, and may diffuse from the gas side outlet port of ECMO leading to the extra-circulatory spread of the SARS-CoV-2 (ECMO infection). Development of a new generation of nanoscale membrane confirmation is proposed for next-generation extracorporeal membrane oxygenator and system with long-term durability is envisaged.

## 1. Introduction

Hollow fiber membrane oxygenator (artificial lung) is used for cardiovascular surgery such as coronary artery bypass surgery, valve replacement, and implantable artificial heart surgery. It is also incorporated into the extracorporeal membrane oxygenation (ECMO) system [1,2,3,4,5,6,7,8], which treats severe acute respiratory distress syndrome due to the novel coronavirus disease 2019 (COVID-19). The extracorporeal membrane oxygenator (ECMO) is said to be the “last stronghold” for critically ill patients with COVID-19 [1].

The most serious dysfunctions in ECMO are the increased excessive pressure drop in blood flow path, plasma leakage, and decrease in gas exchange rate [4,5,6,7,8]. In addition, as blood coagulation and thrombus are generated in severely ill patients with COVID-19 [9], incidents have been reported in which a blood flow path of membrane oxygenator is easily clogged in ECMO treatment. There is a concern that the treatment for critically ill COVID-19 patients causes more serious excessive pressure drop.

On the other hand, plasma leakage frequently occurs in extracorporeal membrane oxygenation (ECMO) when used in respiratory support therapy [4,5], which requires more time than when being used in cardiovascular surgery. When the ECMO membrane pores are in contact with blood for a long time, the hydrophobicity of the membrane is gradually impaired and the membrane becomes hydrophilic, the air in the pores is replaced with plasma, and plasma leaks to the gas side. If plasma leakage occurs, not only is the gas exchange efficiency lowered, but the water balance of the patient is also disturbed and, in the worst case, the patient may be in a critical condition [3]. Even in the case of such an incident, at present, an operator such as a clinical engineer takes measures such as replacing it with an unused membrane oxygenator before an accident occurs.

It has been reported that plasma components in the blood leak into the lumen of hollow fiber membrane (plasma leakage) and yellow foam leaks from the gas outlet port of a membrane oxygenator during an ECMO treatment for critically ill patients with COVID-19 [2]. There is a concern that SARS-CoV-2 in plasma may permeate through the pore in the membrane and diffuse as an aerosol from the gas outlet port, which is still one of the issues in operating ECMO.

The pore structure of the gas exchange membrane for ECMO and the mechanism of plasma leakage have been studied so far [3,4,5]. However, the membrane and pore structures of the recent ECMO membranes have not been clarified, and the permeability of solutes such as viruses through the membrane has not been investigated.

The objective of this study is to clarify the pore structure of ECMO membranes by using our approach for analyzing membrane pore structures by scanning probe microscope (SPM) and field emission scanning electron microscope (FE-SEM) [10,11,12,13,14,15,16]. Then, the permeability of SARS-CoV-2 through the membrane is evaluated using the steric exclusion model and hindered diffusion model, which are simple permeation theories in membrane science. We suggest the development of a new generation of nanoscale membrane confirmation to prevent extra-circulatory spread of the SARS-CoV-2.

## 2. Material and Methods

### 2.1. Hollow Fiber Membrane for Extracorporeal Membrane Oxygenator

The samples studied were commercially available extracorporeal membrane oxygenators, which are typical in Japan. These are the outside blood flow oxygenators. They contribute to ECMO systems in cardiovascular surgery and in the treatment of severe acute respiratory distress syndrome, both in Japan and around the world. Table 1 shows the technical data of the samples.

oxia^®^ ACF (JMS Co., Ltd., Hiroshima, Japan, Sample A) equips a hollow fiber membrane made of polypropylene and is approved as Extracorporeal Membrane Oxygenator (ECMO) in Japan. The MERA NHP^®^ (SENKO MEDICAL INSTRUMENT Mfg. Co., Ltd., Tokyo, Japan, Sample B) is a hollow fiber membrane made of polypropylene coated with a silicone layer on the outer surface. Sample B is approved as ECMO and ECMO for assisting respiration. BIOCUBE^®^ (Nipro Co., Ltd., Tokyo, Japan, Sample C) is a polymethylpentene (PMP) membrane. This is the first PMP membrane for artificial lung worldwide. Sample C is approved as ECMO and ECMO for assisting respiration. The manufacturing approval standard (requirement) [17] of ECMO for assisting respiration is that “the membrane characteristics of a silicone membrane or a special polyolefin membrane can prevent plasma leakage”.

### 2.2. Observation of Three-Dimensional Tortuous Pore Using Scanning Probe Microscope (SPM) System

The method reported in our previous study [15,16] was employed to observe the inner and outer surfaces of hollow fiber membranes.

A sample, as shown in Figure 1, was prepared in order to observe the inner surface of the hollow fiber membrane (dry). A flat surface without curvature was created to improve the accuracy of the image. When observing the outer surface of the hollow fiber, a sample having a low sample height was prepared and a flat surface was observed. Three or more samples were observed, and a representative image was shown in the result.

### 2.3. Observation of Three-Dimensional Tortuous Pores Using Field Emission Scanning Electron Microscope (FE-SEM)

For a comparative verification against the SPM, followed by the design validation, we used a FE-SEM (JSM-7610F, Jeol Ltd., Tokyo, Japan) to observe the inner and outer surfaces of the hollow fiber membranes at an accelerating voltage of 1.5 kV, a working distance of 4.5 mm, and an emission current of 47.2 μA [16]. No conductive treatment with Au or C was applied. Pore diameters were measured in observation fields of a magnification of 100,000 in size by the analysis of digital imagery utilizing ImageJ software (Naional Institute of Mental Health, Bethesda, MD, USA).

### 2.4. Validation of SARS-CoV-2 Permeability Using the Steric Exclusion Model and Hindered Diffusion Model

An attempt was made to quantify the membrane permeability for the SARS-CoV-2 using the steric exclusion model and hindered diffusion model [19].

The diffusive coefficient of SARS-CoV-2 in water was calculated by Equation (1) expressing the Stokes–Einstein relationship.
(1)$$D_{AB}=\frac{RT}{6\pi \mu aN_{A}}$$
where *R* is the ideal gas constant, *T* is the absolute temperature, a is the solute radius, *N_A_* is Avogadoro’s number (6.02 × 10^23^ mol^−1^), and μ is the solution viscosity.

The diffusive coefficient of SARS-CoV-2 in plasma at 37 °C was calculated in consideration of the increase in plasma viscosity relative to the viscosity of water.
(2)$$D_{plasma}=D_{AB}\times \left(\frac{0.697cP}{1.19cP}\right)=0.586D_{AB}$$
where the plasma temperature was set to 37 °C assuming extracorporeal circulation.

Therefore, the fraction of the pore cross-sectional area through which the solute penetrates is given by Equation (3).
(3)$$K=\frac{\pi {\left(r-a\right)}^{2}}{\pi r^{2}}={\left(1-\frac{a}{r}\right)}^{2}=\frac{C_{A\underline{}pore}}{C_{A\underline{}bulk}}$$
where *K* is known as the solute partition coefficient, *a* is molecular radius, *r* is pore radius, and *C_A_* is solute concentration.

Mass transfer flux in a pore is given by Equation (4) from the Fick’s first law applicable to a dilute solution.
(4)$$N_{Ax}^{pore}=-D_{pore}\frac{dC_{A\underline{}pore}}{dx}=-D_{pore}\frac{C_{A\underline{}pore}^{x=\bar{L}}-C_{A\underline{}pore}^{x=0}}{\bar{L}}$$
where $D_{pore}$ is diffusive coefficient in a pore, $\bar{L}$ is membrane thickness, and $C_{A\underline{}pore}$ is solute concentration in a pore.

The solute concentrations in the pore at the inlet and outlet of the pore are related with the bulk solute concentrations just outside the pore by the partition coefficient. Therefore, Equation (4) is expressed as Equation (5).
(5)$$N_{Ax}^{pore}=-D_{pore}\frac{dC_{A\underline{}pore}}{dx}=-D_{pore}K\frac{C_{A\underline{}bilk}^{x=\bar{L}}-C_{A\underline{}bulk}^{x=0}}{\bar{L}}$$

The tortuous nature of the pore is explained by using the tortuosity (τ) for the actual pore length ($\tau \times \bar{L}$).

In addition, as the hindered diffusion affects the permeation of the solute in the pores, the value of $D_{pore}$ is smaller than the value of $D_{AB}$ due to the parameter for the hindered diffusion ($\omega_{r}$). The term for hindered diffusion ($\omega_{r}$) depends on the ratio of the solute radius to the pore radius (a/r). Therefore, Equation (5) can be expressed as Equation (6).
(6)$$N_{Ax}^{pore}=-D_{pore}K\frac{C_{A\underline{}bulk}^{x=\bar{L}}-C_{A\underline{}bulk}^{x=0}}{\bar{L}}=-D_{AB}\left(\frac{K\omega_{r}}{\tau }\right)\frac{C_{A\underline{}bulk}^{x=\bar{L}}-C_{A\underline{}bulk}^{x=0}}{\bar{L}}$$

Equation (6) represents the mass transfer flux in a pore of a membrane in terms of the measurable bulk solute concentrations at the surface on both sides of the membrane. From Equation (6), the solute diffusive coefficient in the pore ($D_{pore}$) and the solute diffusive coefficient across a membrane pore relative to the bulk solute concentration ($D_{m}$) are defined as follows.
(7)$$D_{pore}=D_{AB}\frac{\omega_{r}}{\tau }$$
(8)$$D_{m}=KD_{pore}=D_{AB}\left(\frac{K\omega_{r}}{\tau }\right)$$

From previous studies [20,21], $K\omega_{r}$ is only a function of λ=a/r. This is given by Equation (9).
(9)$$K\omega_{r}={\left(1-\frac{a}{r}\right)}^{2}\left[1-2.1\left(\frac{a}{r}\right)+2.09{\left(\frac{a}{r}\right)}^{3}-0.95{\left(\frac{a}{r}\right)}^{5}\right]$$

The term ${\left(1-\frac{a}{r}\right)}^{2}$ on the right-hand side of Equation (9) is the partition coefficient (K) by Equation (3). The term on the right-hand side of Equation (9) is the hindered diffusion parameter ($\omega_{r}$) by Equation (10).
(10)$$\omega_{r}=\left[1-2.1\left(\frac{a}{r}\right)+2.09{\left(\frac{a}{r}\right)}^{3}-0.95{\left(\frac{a}{r}\right)}^{5}\right]$$

$\omega_{r}$ represents the increased hydrodynamic drag in the pore comparable in size to that of the solute.

## 3. Results

### 3.1. SPM Observations of Tortuous Pore Structures of ECMO Membranes

Figure 2a,b show the results of the observations of oxia^®^ (dry, sample A), (a) the surface of inner lumen of the capillary membranes, and (b) the outer surface of the capillary membranes. In addition, (1), (2), and (3) show the scanning areas of 2000 nm × 2000 nm, 500 nm × 500 nm, and 200 nm × 200 nm, respectively. The image of (3) shows magnified images of the pores enclosed by the blue square in (2), and the image of (2) shows magnified images of the pores enclosed by the blue square in (1). The color gradient bar at the bottom represents the scale in the Z-direction of the image.

On the surface of the hollow fiber membrane of sample A, the unique pore structure stretched in an elliptical shape in the longitudinal direction of the hollow fiber membrane was observed.

Images for the pore structures of ECMO membranes were obtained using SPM, but the images were unclear. Compared with the images of hemoconcentrator membranes and hemodialysis membranes (polyether sulfone) [15,16] in our previous studies, we could not observe clear three-dimensional tortuous pore structures of membranes. Therefore, we decided that an analysis by another approach was necessary.

### 3.2. FE-SEM Observations of Tortuous Pore Structures of ECMO Membranes

Figure 3 is FE-SEM images of oxia^®^ (dry), (a) the surface of inner lumen of the capillary membranes, (b) the outer surface of the capillary membranes: (1) ×30,000; (2) ×50,000; and (3) ×100,000, respectively. The image of (3) shows magnified images of the pores enclosed by the blue square in (2), and the image of (2) shows magnified images of the pores enclosed by the blue square in (1).

On the inner and outer surfaces of the sample A, long elliptical pores were observed in the longitudinal direction of the hollow fiber membrane. The higher the magnification of the image, the deeper the pores could be confirmed, and the three-dimensional tortuous pore structure was confirmed. When polypropylene was stretched by the stretching method to form a hollow fiber membrane, the pores were also stretched in the longitudinal direction of the membrane. The pores on the outer surface were smaller than those on the inner surface. Compared with the SPM images shown in 3.1, the FE-SEM gave very clear images, and the unique pore structure of sample A was observed. For this reason, unlike our previous studies [15,16], FE-SEM is more useful for pore structure analysis of polypropylene ECMO membranes.

Figure 4 is FE-SEM images of MERA NHP^®^ (dry), (a) the surface of inner lumen of the capillary membranes and (b) the outer surface of the capillary membranes.

Sample B is also made of polypropylene, and the pores on the inner surface of the membrane are stretched in the longitudinal direction of the membrane. On the other hand, as the outer surface of the membrane is coated with thin silicone layer, the pore structure is not confirmed. From these, sample B is a polypropylene membrane coated with silicone layer, and in the outside blood flow membrane oxygenator, the silicone layer comes into direct contact with blood. However, as shown in Figure 5, a structure in which the silicone layer was peeled off was partly confirmed on the outer surface of the membrane.

Figure 6 is FE-SEM images of BIOCUBE^®^ (dry), (a) the surface of inner lumen of the capillary membranes and (b) the outer surface of the capillary membranes.

Sample C is a PMP membrane, and the unique pore structure is different from Figure 3 and Figure 4. The inner surface of the membrane has the unique mountain-range structure that includes pores. The outer surface of the membrane is highly porous compared to the inner surface of the membrane. The three-dimensional unevenness of the outer surface of the membrane is considerably larger than the inner surface, but in principle for the FE-SEM, it is difficult to obtain three-dimensional information using the FE-SEM.

In addition, Figure 7 is the image of a different sample taken from the same device. The pore structure on the outer surface of Figure 7 is completely different from that of Figure 6. Many samples had structures similar to the image in Figure 7, but some had structure similar to Figure 6. This is as the skin layer as shown in Figure 7 is formed during the formation (melt-spinning method) of the PMP membrane.

As described above, in this study, the unique pore structures of ECMO membranes, which are commonly used in Japan and worldwide, are clarified in detail for the first time using FE-SEM. In particular, each membrane has completely different anisotropic pore structure on the inner and outer surfaces of the membrane. Extracorporeal membrane oxygenator, which is the outside blood flow type, is the mainstream of membrane oxygenator [22,23,24,25,26,27]. Therefore, it is necessary to appropriately design the pore structure on the outer surface of the membrane that comes into direct contact with blood and the pore structure on the inner surface of the membrane that comes into contact with gas. For this purpose, it is important to control the anisotropic structure of the cross-section of the membrane.

### 3.3. Measurement of Pore Diameter and Pore Diameter Distribution and Evaluation of SARS-CoV-2 Permeability

For a pore diameter measurement, pores that were measured in observation fields of a magnification of 100,000 in size were analyzed. As none of the pores were true circles, the major and minor axis of the pores were measured through a line length analysis. Figure 8 shows the distributions of pore diameters.

Table 2 shows the values for the pore diameter of the ECMO membranes. The pore diameter on the outer surface of sample B was described as a reference as there were only five sets of data that could be observed after the silicone layer was peeled off. Furthermore, the partition coefficient and intramembrane diffusion coefficient of SARS-CoV-2 calculated using the steric exclusion model and the hindered diffusion model described in 2.4 are also shown in Table 2.

In Figure 8 and Table 2, the pore diameters on the outer surfaces of the membranes are smaller than those on the inner surfaces of the membranes. It is necessary to verify what kind of spinning process creates the unique pore structure of each membrane.

The diameter of SARS-CoV-2 is said to be 50–200 nm [28]. Table 2 shows the partition coefficient and the intramembrane diffusive coefficient calculated with the diameter of SARS-CoV-2 at 50 nm and 80 nm which were smaller than the pore diameter. The partition coefficients are greater than 0, intramembrane diffusion coefficients of SARS-CoV-2 are 8.9 × 10^−14^–7.2 × 10^−13^ m^2^/s. Therefore, when a plasma leakage occurs in an extracorporeal membrane oxygenator, SARS-CoV-2 also permeates through the pores of the membrane with the filtration flow of the plasma from the outside to the inner lumen of the membrane. The risk of SARS-CoV-2 permeation in Sample B and Sample C were lower than that of Sample A. The risk of SARS-CoV-2 permeation is completely different due to each anisotropic pore structure, and certainly chemical property. Glucose diffusion coefficient in water calculated by this model is 4.7 × 10^−10^ m^2^/s, while the value in the literature is 9.3 × 10^−10^ m^2^/s. From these data, the credibility of the values calculated by the model is not perfect, but it is reasonably sufficient. The transfer rate of SARS-CoV-2 in the membrane is about 1/1000 of the transfer rate of glucose in water.

## 4. Discussion

### 4.1. ECMO Infection and Usefulness of Theoretically Validating SARS-CoV-2 Permeation through Membrane

Serious dysfunctions in extracorporeal membrane oxygenator are excessive pressure drop in blood flow path due to blood coagulation and thrombosis, and plasma leakage [4,5,6,7,8]. Plasma leakage occurs in extracorporeal membrane oxygenation over a longer term. When a plasma leakage occurs, not only is the gas exchange efficiency is lowered, but the water balance of the patient is also disturbed, and in the worst case, there is a concern that the patient may be in critical condition.

During the current COVID-19 pandemic, it has been reported that the plasma components from the gas outlet port of the membrane oxygenator became yellow foam and positive PCR test results were obtained from the gas outlet port of the membrane oxygenator [2,29]. This is due to the fact that SARS-CoV-2 in plasma may permeate the hollow fiber membrane and aerosol diffusion occurs from the gas outlet port [29]. Therefore, in ECMO support for COVID-19 patients, it must be recognized that there is the risk of ECMO infection due to extra-circulatory spread of the SARS-CoV-2. Moreover, there is a concern that, if the blood coagulation and thrombus are generated in severely ill patients with COVID-19 [9] and they cause a more serious excessive pressure drop, then the transmembrane pressure (TMP) is increased, and plasma and virus easily permeate through the membrane. As the inner lumen of hollow fiber is filled with gas, the TMP is larger than that of the dialyzer in which the inner lumen of hollow fiber is filled with blood. With these concerns, medical staff always use N95 masks, gowns, caps, and face shields to prevent infection with SARS-CoV-2 during ECMO support. Medical staff are burdened by serious pandemic. However, there was no direct evidence of how such a SARS-CoV-2 infection phenomenon occurred. In particular, there is the need to verify the risk of SARS-CoV-2 leakage through PMP membrane and silicone-coated membrane, and to examine the damaged membranes [29].

Therefore, in this study, from the viewpoint of membrane science, we analyzed the recent common ECMO membranes used in Japan and worldwide. The precise pore structure of the membranes was elucidated by direct microscopic observation using FE-SEM. The pore structure of the hollow fiber membrane is not homogeneous but asymmetric. As the pore structures on the inside and the outside of the membrane are different, if the pores on the outside of the membrane are highly microporous, SARS-CoV-2 penetrates from the outside to the inside of the hollow fiber lumen. Here, we find that SARS-CoV-2 may permeate the membrane, transfer from the patient’s blood to the gas side, and diffuse from the gas side outlet port of ECMO. Even in the case of a membrane that suppresses plasma leakage using a silicone-coated membrane such as sample B, plasma leakage may occur due to silicone layer peeled off. In addition, when plasma does not permeate the silicone layer, but gas (water vapor) permeates the silicone layer. Furthermore, condensation (wet lung) occurs in the gas side (inner lumen of hollow fiber) due to a change in the temperature of the gas side, and porous pores on inner lumen are filled with water. In these cases, SARS-CoV-2 is also likely to permeate the membrane wall, even if plasma leakage does not occur. These increase the risk of ECMO infection caused by extra-circulatory spread of the SARS-CoV-2. Additionally, the risk of ECMO infection may be quantitatively evaluated from the viewpoint of membrane science.

In terms of regulatory approval, a PMP membrane with a dense outer layer has been approved for 30 days usage for ECMO (CE marking), whether the plasma leakage is completely prevented requires further study [30]. The FDA has urgently approved the use of ECMO for up to 15 days. In Japan, ECMO is approved for use up to 6 h only. The package inserts for the three samples in this study also state that plasma leakage may occur.

### 4.2. Optimal Design of Asymmetrical Pore Structure of ECMO Membrane

From these perspectives, it is necessary to appropriately design the pore structure on the outer side of the membrane that comes into direct contact with blood as well as the pore structure on the inner side of the membrane that comes into contact with gas. It is important to control the anisotropic structure of the membrane cross-section.

We also focus on the fouling during ECMO treatment [16]. As a next-generation membrane with long-term durability, fouling does not proceed easily, and SARS-CoV-2 does not easily penetrate to prevent extra-circulatory spread of the SARS-CoV-2. These three membranes in this study are contributing to cardiovascular surgery and support for severe acute respiratory distress syndrome, both in Japan and worldwide. Plasma leakage has always been an issue, and its importance is once again recognized due to the current COVID-19 pandemic. In the future, it is necessary to develop next-generation membranes and systems with long-term durability suitable for treatment [31] of COVID-19 critically ill patients. Microstructured hollow fiber membranes that increase the gas exchange surface area were also proposed to improve oxygenator performance [32].

### 4.3. Limitations of Theoretically Validating SARS-CoV-2 Permeation Based on the Membrane Transport Model

In this study, we attempted to evaluate the SARS-CoV-2 permeability using the steric exclusion model and the hindered diffusion model [19] for transport phenomena in membrane. These are fundamental and valuable models for simply calculating the SARS-CoV-2 permeability based on the data listed in Table 2.

However, in this study, data such as the molecular weight distribution, diffusion coefficient in a fluid or quiescent fluid and physical properties of SARS-CoV-2, its affinity with membrane and concentration in plasma have not been available. As soon as such data are accumulated, more detailed validations are feasible. Additionally, there is no actual data on the SARS-CoV-2 permeation, as the suitability of using SARS-CoV-2 in vitro is questionable. To study actual virus permeation, an approach of directly observing virus permeation through membrane wall [33] is also useful.

On the other hand, medical staff are doing rigorous work day by day, so direct evidence is required as described above. Although the output of our study may not provide direct evidence, it provides novel insights to ECMO support of COVID-19 critically ill patients. From our research and other studies [2,29], SARS-CoV-2 is highly likely to permeate the membrane transporting from the patient’s blood to the gas side, and may diffuse from the gas side outlet port of ECMO.

## 5. Conclusions

The precise pore structures of the ECMO membranes were clarified by direct microscopic observation by FE-SEM. The three types of membranes, polypropylene membrane, polypropylene membrane coated with thin silicone layer, and polymethylpentene (PMP) membrane, each have a unique pore structure, and the pore structures on the inner and outer surfaces of the membranes are completely different anisotropic structures. When plasma leakage occurs during long-term prolonged ECMO treatment, SARS-CoV-2 is also likely to permeate through uniquely shaped anisotropic pores with the filtration flow of plasma or wet lung. In this case, SARS-CoV-2 is discharged from the outlet port of the oxygenator gas side, so care must be taken to prevent airborne transmission and aerosol infections of SARS-CoV-2. The risk of SARS-CoV-2 permeation in polypropylene membrane coated with thin silicone layer (Sample B) and PMP membrane (Sample C) was lower. At the time of this current COVID-19 pandemic, the risk of infections on the operators of medical devices is drawing attention. In the future, development of next-generation extracorporeal membrane oxygenator and system with long-term durability is envisaged.

## Figures and Tables

**Figure 1 membranes-11-00529-f001:**
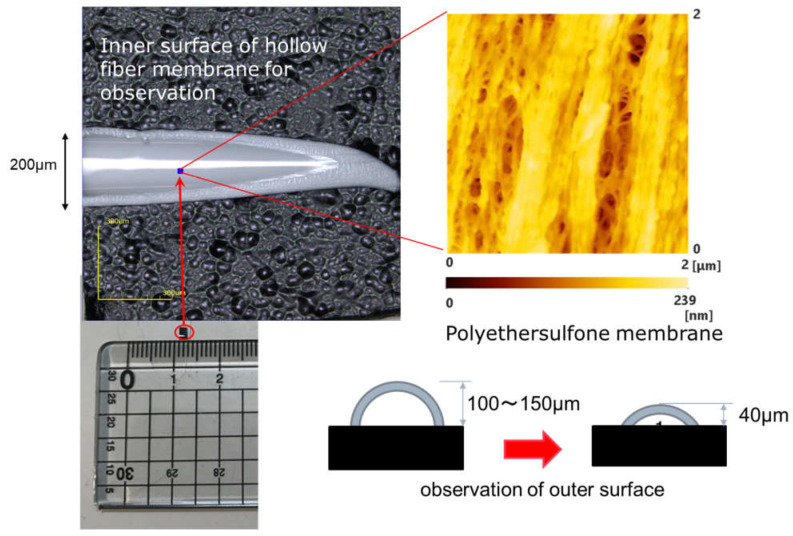
Inner surface of hollow fiber membrane used for observation (LEXT OLS4000, 3D measurement laser microscope, Olympus Co., Ltd., Tokyo, Japan) [15,16].

**Figure 2 membranes-11-00529-f002:**
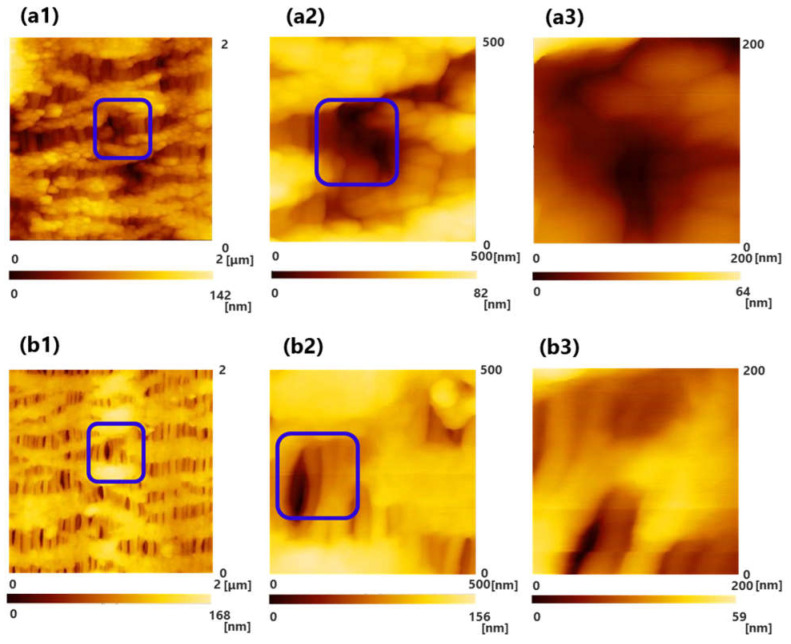
Scanning probe microscopy (SPM) images of oxia^®^ (dry), (**a**) the surface of inner lumen of the capillary membrane, and (**b**) the outer surface of the capillary membrane: (1) 2000 nm × 2000 nm; (2) 500 nm × 500 nm; and (3) 200 nm × 200 nm, respectively. The image of (3) shows magnified images of the pores enclosed by the blue square in (2), and the image of (2) shows magnified images of the pores enclosed by the blue square in (1).

**Figure 3 membranes-11-00529-f003:**
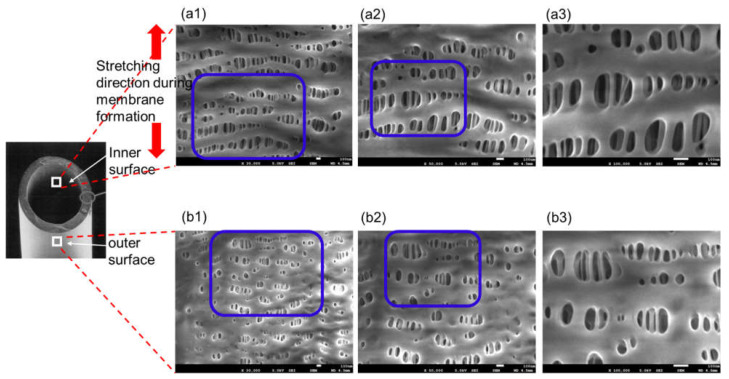
FE-SEM images of oxia^®^ (dry), (**a**) the surface of inner lumen of the capillary membranes, (**b**) the outer surface of the capillary membranes: (1) ×30,000; (2) ×50,000; and (3) ×100,000, respectively. The image of (3) shows magnified images of the pores enclosed by the blue square in (2), and the image of (2) shows magnified images of the pores enclosed by the blue square in (1).

**Figure 4 membranes-11-00529-f004:**
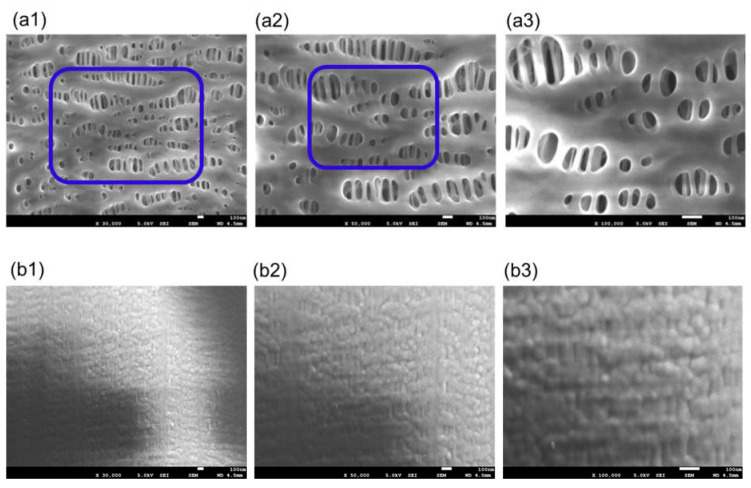
FE-SEM images of MERA NHP^®^ (dry), (**a**) the surface of inner lumen of the capillary membranes, (**b**) the outer surface of the capillary membranes: (1) ×30,000; (2) ×50,000; and (3) ×100,000, respectively. The image of (3) shows magnified images of the pores enclosed by the blue square in (2), and the image of (2) shows magnified images of the pores enclosed by the blue square in (1).

**Figure 5 membranes-11-00529-f005:**
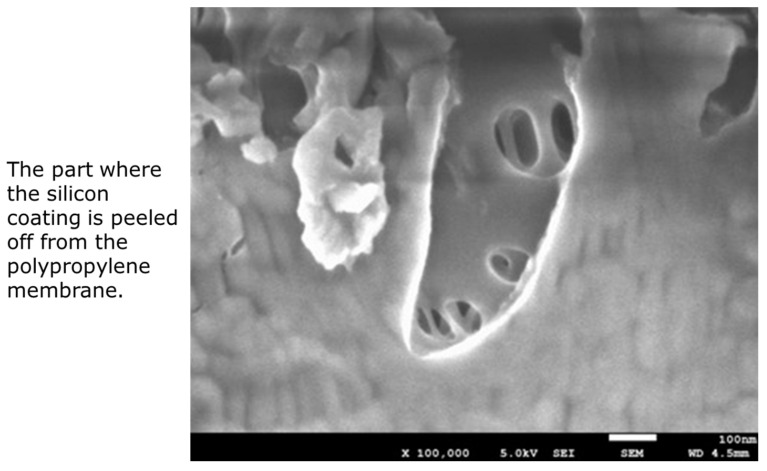
FE-SEM image of MERA NHP^®^ (dry), the outer surface of the capillary membranes; and ×100,000.

**Figure 6 membranes-11-00529-f006:**
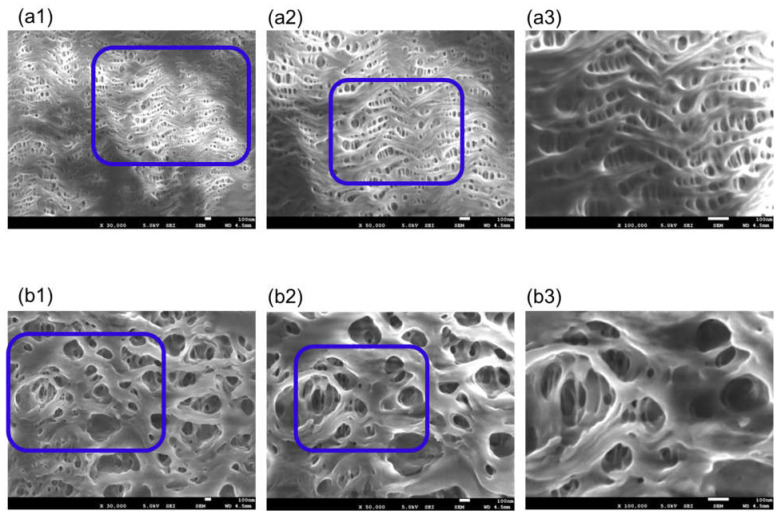
FE-SEM images of BIOCUBE^®^ (dry), (**a**) the surface of inner lumen of the capillary membranes and (**b**) the outer surface of the capillary membranes: (1) ×30,000; (2) ×50,000; and (3) ×100,000, respectively. The image of (3) shows magnified images of the pores enclosed by the blue square in (2), and the image of (2) shows magnified images of the pores enclosed by the blue square in (1).

**Figure 7 membranes-11-00529-f007:**
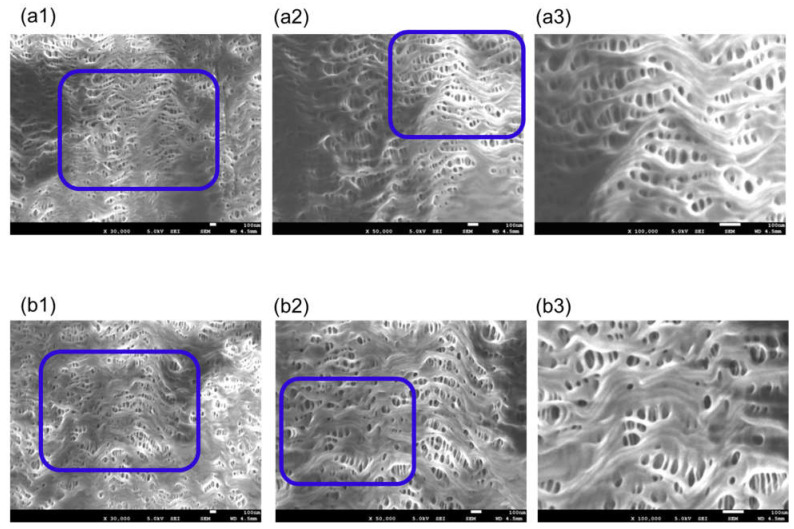
FE-SEM images of BIOCUBE^®^ (dry), (**a**) the surface of inner lumen of the capillary membranes and (**b**) the outer surface of the capillary membranes: (1) ×30,000; (2) ×50,000; and (3) ×100,000, respectively. The image of (3) shows magnified images of the pores enclosed by the blue square in (2), and the image of (2) shows magnified images of the pores enclosed by the blue square in (1).

**Figure 8 membranes-11-00529-f008:**
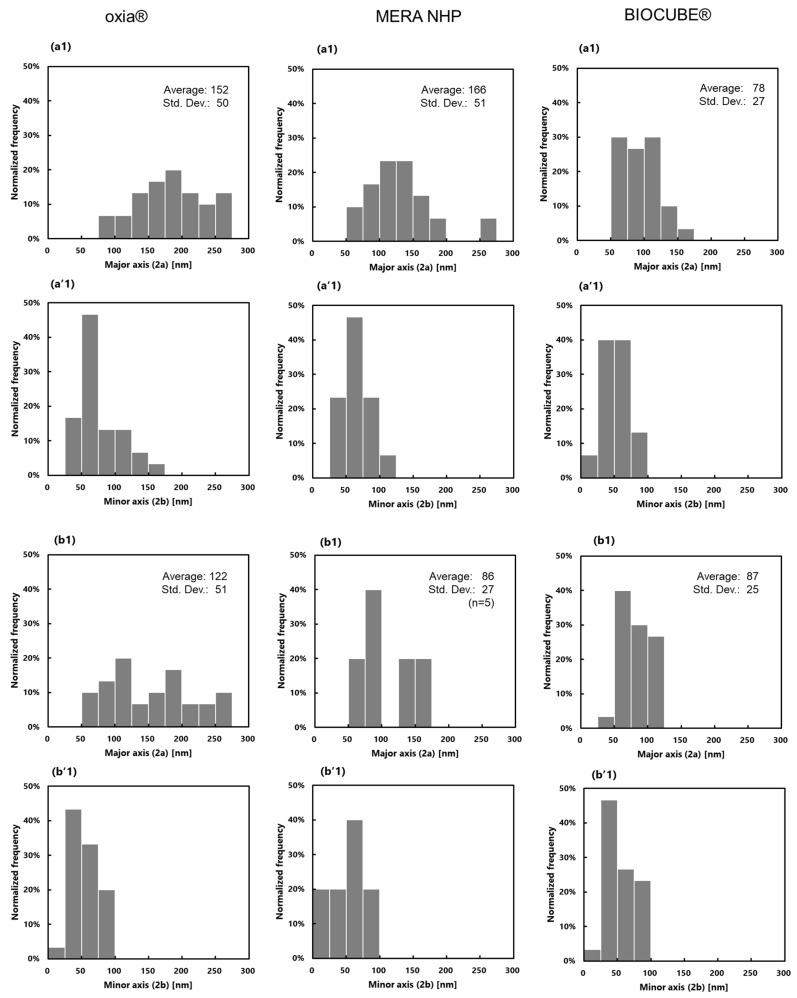
Distribution of the ellipse pore diameter of three membranes (dry), determined via FE-SEM, (**a**) the surface of inner lumen of the capillary membranes and (**b**) the outer surface of the capillary membranes. (**a1**,**b1**) Major axis (2a), (**a’1**,**b’1**) Minor axis (2b).

**Table 1 membranes-11-00529-t001:** Specification of hollow fiber membranes for membrane oxygenators.

Sample	oxia^®^ ACF Sample A	MERA NHP^®^ Exelung TPC HPO-23WH-C Sample B	BIOCUBE^®^ 6000 Sample C
Manufacturer (Manufacturer of membrane)	JMS Co., Ltd., Hiroshima, Japan (3M Co., Ltd., USA)	SENKO MEDICAL INSTRUMENT Mfg. CO., Ltd., Tokyo, Japan	Nipro Co., Ltd., Tokyo, Japan (DIC Co., Ltd., Tokyo, Japan)
Material of hollow fiber membrane	Polypropylene (PP)	Polypropylene (PP), silicone	Polymethylpentene (PMP)
Antithrombogenic material coating for blood flow channel	Poly (2-methacryloyloxyethyl phosphoryl choline) (PMPC)	Heparin compound	Heparin
Inner diameter of lumen [µm] (n = 30)	238 ± 5	246 ± 3	176 ± 6
Membrane thickness [µm] (n = 30)	35 ± 1	27 ± 1	30 ± 2
Air permeability ^(1)^ [s]	35.1 ± 1.4	-	-
Pore structure	asymmetric pore structure	asymmetric pore structure, the outer surface of the membrane is coated with thin silicone layer (thickness 0.2 µm)	asymmetric pore structure
Sterilization method	EOG	EOG	EOG
Insurance coverage classification (in Japan)	extracorporeal membrane oxygenator; ECMO ^(2)^	extracorporeal membrane oxygenator; ECMO ^(2)^ Cardiac ECMO Respiratory ECMO ^(3)^	extracorporeal membrane oxygenator; ECMO ^(2)^ Cardiac ECMO Respiratory ECMO ^(3)^

^(1)^ The gas permeability of the membranes was measured by the Gurley method using an air resistance tester defined in ISO5636-5 [18]. The smaller the value, the larger the gas permeability; ^(2)^ Usage time 6 h; ^(3)^ Usage time 6 h, the membrane characteristics of a silicone membrane or a special polyolefin membrane can prevent plasma leakage.

**Table 2 membranes-11-00529-t002:** Pore diameter of ECMO membranes and SARS-CoV-2 permeability.

Sample	Oxia^®^ ACF Sample A	MERA NHP^®^ Exelung TPC HPO-23WH-C Sample B	BIOCUBE^®^ 6000 Sample C
Tortuous pore diameter of inner surface (nm) n = 50, AVG. ± STD. upper: major axis lower: minor axis	152 ± 50	166 ± 51	78 ± 27
	59 ± 25	53 ± 15	42 ± 13
Tortuous pore diameter of outer surface (nm) n = 50, AVG. ± STD. upper: major axis lower: minor axis	122 ± 51	86 ± 27	77 ± 25
	44 ± 14	40 ± 18 (n = 5)	52 ± 15
Partition coefficient (K) of SARS-CoV-2 [-] SARS-CoV-2 diameter upper: 50 nm lower: 80 nm	0.35	0.18	0.12
	0.12	0.005	0.002
Intramembrane diffusive coefficient (D_m_) of SARS-CoV-2 ^(1^^)^ ^(2)^ (m^2^/s) SARS-CoV-2 diameter upper: 50 nm lower: 80 nm	7.2 × 10^−13^	1.6× 10^−13^	8.9 × 10^−14^
	5.5 × 10^−14^	1.5× 10^−15^	8.4 × 10^−17^

^(1)^ D_m_ was calculated with the tortuousity on the outer surface side of the membrane as 1. It needs to be modified in the future. ^(2)^ Calculated glucose diffusion coefficient 4.7 × 10^−10^ m^2^/s, 9.3 × 10^−10^ m^2^/s (the value in the literature) [19].

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
