# Peer review of "Electron Microscopic Confirmation of Anisotropic Pore Characteristics for ECMO Membranes Theoretically Validating the Risk of SARS-CoV-2 Permeation"

_membranes, 2021, doi:10.3390/membranes11070529_

Round 1

Reviewer 1 Report

Very interesting manuscript, well written, with novel-original content. 

Methods and results adequately presented 

Discussion is focused and clearly signposted key messages for the reader are included

Limitations of the study well presented and conclusions are realistic and reasonable 

Reviewer 2 Report

I have no further comments.

This manuscript is a resubmission of an earlier submission. The following is a list of the peer review reports and author responses from that submission.

Round 1

Reviewer 1 Report

The authors implement an analysis of the membrane structure of various membranes, based on their earlier works, and implement two basic well known membrane transport model to estimate the transport of covid -19 across the membranes.

Although I find the analysis of the pore morphology interesting, the results are actually quite predicable. Leakage of plasma across the oxygenator membrane with such large pores could lead to leakage of blood components including viruses as covid. The authors should include actual transport data to actually validate their findings.

Based on the findings and estimations they authors conclude that real conclusions cannot be made about the covid transport unless transport data are available which actually, I think, proves my earlier point.

The discussion section is unfortunately of low level, too short with no important new insights rather the known suggestions to change the pore shape here and there and to avoid fouling.

The English needs major revision often leading to misleading text. If one reads the abstract and discussion, he/she might think that the transport across the membranes is actually measured and the model is validated. 

Reviewer 2 Report

In this manuscript the authors evaluate the pore size and structure of 3 ECMO membranes: polypropylene, coated polypropylene and polymethylpentene using scanning electron microscopy. They also perform a mathematical exercise to demonstrate that covid 19 passes through the membrane. 

By far the strongest part of this paper is the pore size evaluation of the 3 tested membranes. This is informative and adds to our knowledge.

I have a problem with the Einstein Stokes equation which the authors use. Basically this equatrion assumes that the particles are in a quiescent fluid. This is typically not the case during ECMO. How do the authors justify their approach?

In general, the authors overstate the importance of their calculations throughout the article.

The references need a careful eye. doi links are not correct, a number of references cannot be found using Pubmed, the presentation of the references is not uniform and most important a number of references is only available in Japanese which prohibits any check.

Although I appreciate that English is not the native language of the authors, and I would certainly not be able to write a manuscript in Japanese, I recommend professional editing of the text.

Abstract: Should be rewritten and structured with research question, results and interpretation. Please stick to your results.

Would it not be better to omit completely the SPM observations in paragraph 3.1? These pictures do not seem to add anything. To me it is confusing and the SEM pictures are both beautiful and informative.

Most researchers agree that the size of Covid 19 is between 50 and 150 um. Would it not be more appropriate to give a range in table 2, instead of only the calculation of the author’s assumption of 50 um?

ln 19 Please change reveals in suggests 

ln 45-50 distracts and does not add to the message of this manuscript

ln 56 may benefit from a reference

ln 72 Please explain SPM and FE-SEM

ln 321-334 should be formulated more carefully. The authors don’t demonstrate passage of Covid through the membrane, they use calculations and assumptions. The authors are aware of this given their restrictions in ln 350-355.

ln 366 Please state what membrane could be superior. 

ln 367 please omit highly. You overstate your results.

Reviewer 3 Report

Authors compared 3 membrane oxygenators about the pore size and saw if there is a risk of SARS-Cov-2 permeation using EM.

Major criticisms:

This is a theoretical conclusion that SARS-Cov-2 permeates the  membrane merely based on the pore size and the size of the virus. There is no direct data. Also, there are no data that it will pose a risk of infection for ECMO operators. How about other viruses?

How did those equations were used in Results?

Do flow rate, viscosity, hematocrit, and shear force affect permeability?

Lines 294-305 It should not be in results.

Minor criticisms:

ECMO membrane is nor right terminology. It should be membrane oxygenator

There is spell out of ECMO too many times.

Line 42 I don't understand blood coagulation and thrombus mean.

Spell out SPM ad FE-SEM at their first appearance in the text.

Line 90 What is the difference of ECMO and ECMO for assisting respiration? Does it mean VA ECMO and VV ECMO?

Does MERA NHP require ®?

Lines 176-180 are unnecessary.